# The Effect of *Poria cocos* Polysaccharide PCP-1C on M1 Macrophage Polarization via the Notch Signaling Pathway

**DOI:** 10.3390/molecules28052140

**Published:** 2023-02-24

**Authors:** Xuerui Hu, Bangzhen Hong, Xiaoxiao Shan, Yue Cheng, Daiyin Peng, Rongfeng Hu, Lei Wang, Weidong Chen

**Affiliations:** 1School of Pharmacy, Anhui University of Chinese Medicine, Hefei 230001, China; 2Anhui Province Key Laboratory of Chinese Medicinal Formula, Hefei 230001, China; 3Institute of Traditional Chinese Medicine Resource, Anhui University of Chinese Medicine, Hefei 230001, China; 4Anhui Province Key Laboratory of Pharmaceutical Preparation Technology and Application, Anhui University of Chinese Medicine, Hefei 230001, China; 5Key Laboratory of Xin’an Medicine Ministry Education, Anhui University of Chinese Medicine, Hefei 230001, China

**Keywords:** *Poria cocos*, homogeneous polysaccharide, macrophage polarization, Notch signaling pathway

## Abstract

The homogeneous galactoglucan PCP-1C extracted from *Poria cocos* sclerotium has multiple biological activities. The present study demonstrated the effect of PCP-1C on the polarization of RAW 264.7 macrophages and the underlying molecular mechanism. Scanning electron microscopy showed that PCP-1C is a detrital-shaped polysaccharide with fish-scale patterns on the surface, with a high sugar content. The ELISA assay, qRT-PCR assay, and flow cytometry assay showed that the presence of PCP-1C could induce higher expression of M1 markers, including tumor necrosis factor-alpha (TNF-α), interleukin-6 (IL-6), and interleukin-12 (IL-12), when compared with the control and the LPS group, and it caused a decrease in the level of interleukin-10 (IL-10), which is the marker for M2 macrophages. At the same time, PCP-1C induces an increase in the CD86 (an M1 marker)/CD206 (an M2 marker) ratio. The results of the Western blot assay showed that PCP-1C induced activation of the Notch signaling pathway in macrophages. Notch1, ligand Jagged1, and Hes1 were all up-regulated with the incubation of PCP-1C. These results indicate that the homogeneous *Poria cocos* polysaccharide PCP-1C improves M1 macrophage polarization through the Notch signaling pathway.

## 1. Introduction

*Poria cocos* are the dried sclerotia from *Poria cocos* (Schw.) Wolf, a fungus of the Polyporaceae family, which has an essential position in traditional Chinese medicinal applications. It has been confirmed to have diuretic, sedative, anti-inflammatory, and immunomodulatory effects [1]. Extracted *Poria cocos* polysaccharides (PCP) are some of the significant effective ingredients of *Poria cocos*. It not only has minor toxicity and side effects, but it also possesses a broad spectrum of pharmacological activities [2], such as being hepatoprotective [3], anti-inflammatory [4], anti-tumor [5], immunomodulatory [6], etc. In our previous study, the natural active ingredient, homogeneous polysaccharide PCP-1C, a galactoglucan-containing mannose mainly composed of 1,6-α-D-galactose and α-D-mannose as its repeating unit, isolated and purified from PCP, was confirmed to have excellent hepatoprotective activity (Figure 1) [7]. However, little has been reported about the possibility of PCP’s influence on immune regulation.

The immune system is vital in staying healthy overall and resisting various diseases [5,8]. Macrophages are an indispensable part of adaptive immunity and innate immunity. Their polarized forms, M1 and M2, are important therapeutic targets for cancer therapy [3,9]. The structure of polysaccharides can have a great influence on their biological activities. Polysaccharides containing glucan and/or mannan units can generally be recognized by carbohydrate receptors on macrophages, such as Toll-like receptors, further inducing the polarization of the macrophages [10,11]. In addition, it has been shown that the crude extraction of PCP and its derivative, PCP with 1% sodium carbonate (PCSC), can activate macrophages via the NF-κB pathway [12,13].

Notch signaling is an important regulator of cell development and differentiation. Mammals express four transmembrane Notch receptors (Notch1–4) and five typical transmembrane ligands, including Delta-like (Dll)-1, -3, and -4 and the Jag-type ligands, Jag-1 and -2. The Notch signaling pathway is considered as playing a major role in the regulation of tumors, such as those in breast and thyroid cancer [14,15], etc., and inflammation. It can regulate the phenotype of tissue-resident macrophages in liver inflammation to promote inflammation [16], up-regulate the expression of inflammatory factors in atherosclerosis [17], and induce neuroinflammation [18]. At the same time, the signal strength of the Notch pathway can be precisely regulated by cellular regulatory mechanisms because its signaling relies on stoichiometric interactions between pathway components [19]. Therefore, targeting the Notch pathway may be a more effective way to regulate tumors and inflammation. The Notch signaling pathway in macrophages plays an essential part in their polarization process [20]. Activation of the Notch pathway in macrophages can regulate the polarization of macrophages into the M1-type, prompting the release of larger amounts of the anti-inflammatory cytokine and aggravating the inflammatory response, producing positive effects in tumor treatments [21]. In the Notch signaling pathway, after binding the activated ligand Jagged1 to the Notch1 receptor [22,23], the downstream target genes Hes1 and Deltex are activated, consequently regulating the level of inflammatory factor secretion, such as that of tumor necrosis factor-alpha (TNF-α), interleukin-6 (IL-6), etc., through the NF-κB pathway [24,25,26]. Therefore, the exploration of whether the homogeneous polysaccharide PCP-1C affects the polarization of macrophages through the Notch pathway is of great significance to the study of the regulation of tumor immunity.

The purpose of the present study was to examine the influence of PCP-1C on M1 macrophage polarization and its underlying mechanism. Macrophage-like cells from the RAW264.7 cell line were stimulated to develop M1 polarization via LPS. Meanwhile, with the presence of LPS, PCP-1C was used to indicate the differentiation of the macrophage polarization. We measured the expression of TNF-α, interleukin-6 (IL-6), interleukin-12 (IL-12), interleukin-10 (IL-10), and their messenger RNA (mRNA), as well as macrophage surface markers CD86 and CD206. The determination of the expression levels of related factors, including Notch1, Jagged1, and Hes1, was used to detect whether the Notch signaling pathway was involved in PCP-1C-induced polarization. This study was able to demonstrate that PCP-1C has the possibility to regulate the immune response by affecting macrophage polarization, which will help improve the understanding of the biological and pharmacological activities of PCP-1C.

## 2. Results

### 2.1. The Appearance and Chemical Composition of PCP-1C

PCP-1C is a white powder, as shown in Figure 2A. Scanning electron microscope (SEM) observation revealed that PCP-1C is mostly comprised unfolding, thin, detrital flakes and has a sheet-like, compact structure, with fish-scale patterns on the surface (Figure 2B). The edge of the fish-scale pattern is jagged, and the surface is uneven, but there is no large protrusion when observed under higher magnifications (Figure 2C). The reason for the formation of these patterns might be the composition of the polymer in the polysaccharide. The results of the physical–chemical assessment showed that the sugar content of PCP-1C is as high as 96.97%, almost free of protein and uronic acid, with 0.61% and 0.07% of those substances, respectively. Combined with our previous study [7], our analysis illustrated that the purity of the extracted PCP-1C was relatively high and belonged to a homogeneous polysaccharide (Table 1).

### 2.2. Homogeneity and Molecular Weight

As shown in Figure 3, the profile of PCP-1C exhibited a single symmetrical peak, indicating that it was a homogeneous polysaccharide. The molecular weight of the polysaccharide was determined by a universal calibration curve using polyethylene glycol as a standard. The molecular weight calibration curve was Y = 16.488417 − 0.884319 X^1^ + 0.016204 X^2^ − 0.000079X^3^ (Y = logMw, X = Rt). According to the retention time of PCP-1C, the molecular weight of PCP-1C was determined to be 11.75 kDa.

### 2.3. The Effect of PCP-1C on the Morphology, Viability, and Phagocytosis of RAW264.7

The addition of LPS (2 μg/mL) aimed to induce polarization to the M1 phenotype, which can be considered as an M1 model. PCP-1C (100 μg/mL) was added on the basis of LPS administration with 24 h incubation in order to detect the effect of PCP-1C on the polarization of macrophages. From the results, 2 μg/mL LPS-induced macrophages experienced morphological changes; they polarized into polygonal shapes, and this shape was considered as the M1 phenotype, rather than the round macrophages in the control group (Figure 4A,B). The macrophages treated with 100 μg/mL PCP-1C have a similar morphology to the macrophages in the LPS group, and had a more significant number of deformed cells, with sharper, larger, and longer angles (Figure 4C). The MTT assay was used to detect cell viability. The results are shown in Figure 4D. When compared with the control group, after treatment with different concentrations of PCP-1C, there was no significant inhibitory effect of PCP-1C treatment on cell viability in RAW264.7 cells, but rather it showed a proliferation effect, whereas the activity of the RAW264.7 cells showed a downward trend (*p* < 0.05) with increases in the concentration of PCP-1C, which suggested that the higher concentration of PCP-1C yielded greater cytotoxicity. The neutral red assay was implemented to detect the phagocytic ability of macrophages. The ability of the cells to phagocytose neutral red was significantly improved (*p* < 0.05) with the increase in the PCP-1C concentration, which increased from 121% induced by 0 μg/mL PCP-1C to 255% generated by 200 μg/mL PCP-1C (Figure 4E). These results indicated that PCP-1C has a proliferative effect on RAW264.7 over a range of concentrations, and the phagocytic capacity increased dose-dependently.

### 2.4. The Effects of PCP-1C on the Cytokine Profile of Macrophages

In order to explore the effects of LPS and PCP-1C on macrophages, we first used ELISA to identify the secretion of cytokines. Based on the results presented (Figure 5A–C), the levels of IL-6, IL-12, and TNF-α from the macrophages treated with LPS for 24 h were significantly higher than those from the control group (*p* < 0.05). PCP-1C dose-dependently induced the increase in pro-inflammatory factors stimulated by LPS from 26.10 ± 0.77 μg/mL to 27.83 ± 0.72 μg/mL (IL-6), from 16.65 ± 0.13 μg/mL to 18.23 ± 0.38 μg/mL (IL-12), and from 57.96 ± 1.70 μg/mL to 70.27 ± 3.64 g/mL (TNF-α). Meanwhile, levels of these three factors secreted by cells treated with 100 μg/mL and 200 μg/mL PCP-1C were found to be significantly higher than those in the LPS group (*p* < 0.05). By contrast, the levels of anti-inflammatory factor IL-10 induced by PCP-1C and LPS were significantly lower than 84.98 ± 1.64 μg/mL in the control group and 80.16 ± 1.47 μg/mL in the LPS group, with a decrease from 75.69 ± 2.11 μg/mL to 70.13 ± 1.56 μg/mL, in an obvious dose-dependent manner (Figure 5D) although the amount of IL-10 was higher than the released amount of the above three factors.

The results obtained by PCR analysis were similar to those found via ELISA. As shown in Figure 6A–D, after quantifying the expression of these four cytokines, the expression levels of pro-inflammatory cytokines in the LPS group, including IL-6 (1.49 ± 0.0.49), IL-12 (3.21 ± 1.56), and TNF-α (2.38 ± 0.22), were higher than those in the control group. Incubation with PCP-1C further triggered up-regulation of the transcription of these three factors. With the increase of the PCP-1C concentration, the expression of pro-inflammatory factors also increased (IL-6: from 1.67 ± 0.26 to 4.98 ± 0.40, IL-12: from 6.66 ± 1.60 to 16.52 ± 1.50, and TNF-α: from 2.98 ± 0.22 to 0.24 ± 0.06). Of these, 100 μg/mL and 200 μg/mL PCP-1C had significant effects on the secretion of pro-inflammatory factors (*p* < 0.05).

### 2.5. The Effects of PCP-1C on the Macrophage Polarization Detected by Flow Cytometry

To further address PCP-1C’s effects on macrophages, flow cytometry was carried out to quantitatively analyze the surface markers of the macrophages induced by PCP-1C and LPS. PCP-1C’s effects were then indicated according to the CD86/CD206 expression ratio (Figure 7G). The results showed no significant difference between CD86 and CD206 in the control group, with a ratio of 1.44 ± 0.04. After 1 μg/mL of IL-4 stimulation for 24 h, CD206 expression on RAW264.7 cells experienced an increase, corresponding to a decrease in the CD86/CD206 ratio to 0.327 ± 0.06, which suggested that the RAW264.7 cells polarized to M2 (Figure 7A,F). When administered solely LPS, the expression of CD86 increased (Figure 7B), and the ratio increased to 2.84 ± 0.40. Coupled with the appearance of PCP-1C, CD86/CD206 increased with its concentration on the basis of the LPS group (Figure 7C–E), and the ratio increased to 7.91 ± 1.68. The CD86/CD206 ratios induced by 100 μg/mL and 200 μg/mL PCP-1C were significantly higher than those of the blank and LPS groups (*p* < 0.05). This means that CD86, a marker for M1-type macrophages, has more positive signals than M2 markers. Therefore, based on the detection of ELISA, PCR, and flow cytometry, PCP-1C can increase the pro-inflammatory factors of macrophages, reduce anti-inflammatory factors, and increase the ratio of M1(CD86)/M2(CD206).

### 2.6. PCP-1C Affects M1 Polarization via Notch Signaling Pathway

To examine the mechanism of PCP-1C-induced macrophage polarization, we implemented the Western blot assay to present the variation of Notch signaling molecules (Figure 8A). As shown in Figure 8C, PCP-1C and LPS triggered a dose-dependent up-regulation of the expression of Notch 1. When the PCP-1C was increased to the concentrations of 100 μg/mL and 200 μg/mL, the expression of Notch 1 was significantly up-regulated (*p* < 0.05), indicating that PCP-1C and LPS are strong inducers. In addition, we detected the expression of the ligand Jagged1 and the target gene Hes1 in the Notch signaling pathway, which were expressed as activation markers of the Notch signaling pathway. As shown in Figure 8B,D, treatment with LPS and PCP-1C resulted in the up-regulation of Jagged1 and Hes1 expression. In summary, LPS and PCP-1C stimulated Notch receptor expression in the macrophage-like cell line RAW264.7, further triggering Notch signaling activation, which indicated that the Notch signaling pathway had been activated, and that it has the potential to participate in the immune regulation of PCP-1C.

## 3. Discussion

PCP has many effects on pharmacology, including a positive role in regulating immune function and antitumor treatment [5,27,28]. Studies have demonstrated that PCP can suppress tumor cell growth by regulating Bcl-2/Bax protein [27], and carboxymethyl sulfated PCP can induce necrosis and apoptosis in tumor cells [6]. However, due to the difficulty of extraction, most of the pentachlorophenols currently under study are crude extracts or derivatives. Tang et al. found that the molecular weight of PCP within a certain range has a greater resistance to oxidation and tumors [29], suggesting that homogeneous polysaccharides may have better bioactivity and therapeutic potential. However, the effect of the homogeneous polysaccharide PCP-1C on macrophages had not been studied. In this research, we determined whether or not PCP-1C impacts macrophage polarization with in vitro macrophages as the stimulus object, which can be suggestive of the possible effects of PCP-1C on the immune response.

Macrophages are crucial components of the inflammatory microenvironment [30]. Activated macrophages have anti-inflammatory or pro-inflammatory functions for promoting local tissue damage or cell regeneration. Among them, M1-type macrophages under stress have strong antigen-presentation ability and can secrete a large number of pro-inflammatory cytokines, such as TNF-α, IL-6, and IL-12 [31,32,33], which have killing effects on tumor cells [34,35]. Among these, TNF-α and IL-6 are considered to be helpful for tumoricidal activity [36,37], and IL-12 is very important for regulating the type and duration of immune response [38,39]. M2-type macrophages can produce anti-inflammatory factors, such as IL-10, which can promote tumor cells’ survival, proliferation, and spread [40]. In the present study, we found that the expression of the M1-type macrophage characteristics can be enhanced by PCP-1C-treatment, including the increase in the M1(CD86)/M2(CD206) ratio, the polygonal appearance, and the dose-dependent increase in the secretion of pro-inflammatory cytokines IL-6, TNF-α, and IL-12 [41,42,43]. Conversely, PCP-1C down-regulated the expression of anti-inflammatory factor IL-10. Therefore, PCP-1C has the possibility to participate in the treatment of immune-regulation-related diseases through the polarization of macrophages to the M1 phenotype. The recognition of polysaccharides by macrophages is inseparable from their structure, and polysaccharides containing glucose and mannose are more easily recognized [44]. Extracellular polysaccharides from Auricularia auricular, containing β-glucan similar to PCP-1C, have the ability to polarize M2 macrophages to M1s [45]. The reported maca polysaccharide MC-2 and CMPB90-1, which contain glucose and mannose, have anti-tumor effects and can stimulate macrophage polarization to the M1-type [7,46,47,48]. They have a part of a polysaccharide structure similar to PCP-1C [7], suggesting that PCP-1C has the potential for anti-tumor ability through macrophages, but further research is needed to confirm this.

Regarding the study of the activation mechanism of polarization, identifying macrophages and activating signal transduction pathways through specific receptors/proteins located on the macrophage membrane is an important step [49]. The Notch signaling pathway plays an essential role in macrophage polarization. Several studies have reported that the activation of the Notch signaling pathway has positive effects on tumor treatments [21]. Astragalus polysaccharides can induce the polarization of a macrophage toward the M1 phenotype through the Notch pathway, further leading to the subsequent inhibition of tumor growth [50]. Up-regulating the Notch pathway on macrophages in the Lewis lung carcinoma cell (LCC) model can lead to the repression of tumor growth by enhancing the polarization to the phenotype with anti-tumor activity [51]. Moreover, stimulation of pro-inflammatory factors contributes to the polarization of M1 macrophages and the simultaneous upregulation of Notch pathway molecules, which can activate canonical Notch signaling [52]. When macrophages are exposed to an inflammatory environment, Notch receptor ligands increase in a TLR-4- and NF-κB-dependent manner, leading to Notch proteolysis and activation, which increases the transcription of pro-inflammatory genes, leading to MAPK, Akt, and the activation of the NF-κB pathway [53]. Activation of Notch signaling is associated with macrophage activation and polarization. Therefore, we investigated whether PCP-1C could activate M1 polarization by affecting the Notch signaling pathway. In addition, some reports have indicated that Notch signaling involves the participation of the NF-κB and MAPKs pathway [54,55], that PCP can stimulate the up-regulation of NF-κB and TLR-4 [6,13], and that Notch antagonists prevent the M1 macrophage polarization stimulated by polysaccharides [50]. Meanwhile, our experiments have shown that PCP-1C can induce the activation of the Notch signaling pathway. Therefore, PCP-1C-triggered M1 polarization might occur through a combination of different pathways, including Notch, TLR-4/NF-κB, MAPKs, etc. However, due to the complexity of polarization and the Notch signaling pathway, as well as the differences between the active ingredients contained in PCP and PCP-1C, more comprehensive studies are needed to conclusively demonstrate the involvement of Notch pathway molecules in macrophage-specific subtype polarization.

Although our studies using an in vitro mouse peritoneal macrophage culture model demonstrated the partial pharmacological activity of PCP-1C, further investigations, in vivo animal models, and clinical trials are still needed to evaluate the role of PCP-1C in immune regulation. Furthermore, there are many other pathways and receptors involved in the Notch pathway activation, and further research on the mechanism of macrophage polarization is needed.

## 4. Materials and Methods

### 4.1. Materials and Reagents

The *Poria cocos* sclerotium was obtained from Guangyintang Traditional Chinese Medicine Company (Anhui, China). Murine macrophage cell lines RAW 264.7 were gained from the Chinese Academy of Sciences. Beyotime Biotechnology (Shanghai, China) was the source of the dimethyl sulfoxide (DMSO), neutral red solution, and MTT powder. Lipopolysaccharide (LPS) was offered by Sigma-Aldrich, Shanghai, China). ELISA kits for the quantification of TNF-α, IL-10, IL-6, and IL-12, and the BCA kit were obtained from Jianglai Biotech Co. (Shanghai, China). The SPARKscript II RT Plus Kit for reverse transcription and the 2×SYBR Green qPCR Mix for amplification were bought from Cisco Biotech Co. (Binzhou, China). Primers were purchased from Sangon Biotech Co. (Shanghai, China). FITC anti-CD206 and PE anti-CD86 were supplied by BioLegend (San Diego, CA, USA). Primary antibodies against Notch1, Hes1, β-actin (Zhengneng Bio Co., Chengdu, China), Jagged1 (Boaosen Bio Co., Beijing, China), respectively, were used. All other reagents mentioned in this study were of analytical grade.

### 4.2. Morphological Observation and Preparation of PCP-1C

#### 4.2.1. Preparation of PCP-1C

The extraction method of PCP was in accordance with previous research methods [7]. Briefly, after the dried *Poria cocos* powder was mixed with ultrapure water, adding ethanol to precipitate, the protein in the sediment was removed, and it was reconstituted. The resulting solution was dialyzed, concentrated, and dried subsequently to obtain PCP. In the next step, the PCP was purified by a DEAE cellulose 52 column (1.6 × 60 cm) and eluted with a 0–1.0 M NaCl gradient. Further elution with ultrapure water was operated on a Sephacryl S-500 chromatographic column (1.6 × 70 cm). The third component was PCP-1C.

#### 4.2.2. Scanning Electron Microscope Observation and Chemical Composition Detection

A scanning electron microscope (SEM, SU1510, Hitachi High Technologies, Tokyo, Japan) was used for the observation of the morphology of PCP-1C. The dried sample powder was mounted on the sample stage with conductive double-sided tape, and gold was sprayed under vacuum conditions. Glucose was used as the standard, and the carbohydrate content was measured at 490 nm via the phenol-sulfuric acid method. With galacturonic acid as the standard, the total uronic acid content was quantitatively determined at 520 nm by the m-hydroxybiphenyl colorimetric method. With bovine serum albumin powder as the standard, the protein content was revealed to be 595 nm via the Coomassie brilliant blue method [7].

#### 4.2.3. Homogeneity and Molecular Weight

The homogeneity and molecular weight of the PCP-1C were determined by high-performance gel permeation chromatography (HPGPC). The sample was dissolved in the aqueous phase (50 mmol/L sodium nitrate, 0.02% sodium azide), filtered through a 0.22 μm microporous filter, and detected with an Agilent 1260 system at a flow rate of 0.5 mL/min (35 °C). The injection volume was 100 μL (10 mg/mL). The homogeneity of the sample was judged by the shape and width of the HPGPC chromatographic peaks.

### 4.3. The Stimulation to Polarization of the Macrophages by PCP-1C

#### 4.3.1. Cell Culture

RAW264. 7 cells were purchased from the Chinese Academy of Sciences. The cells were cultured in a 6-well plate (1 × 10^6^ per well) and divided into a control group (RPMI-1640 medium), LPS group (2 μg/mL), and LPS administration groups (2 μg/mL) with different concentrations of PCP-1C (50, 100, and 200 μg/mL). The cells in each group had three parallel holes. Cell attachment was allowed prior to incubation with the corresponding test substance for 24 h.

#### 4.3.2. Cell Viability

The colorimetric MTT assay was used to evaluate the viability of the PCP-1C-induced RAW264.7 cells. After the cells adhered to the wall, we used 2 μg/mL LPS and 0, 50, 100, 200 μg/mL PCP-1C, or blank culture medium to continue culturing for 24 h. A quantity of 20 μL of MTT (5 mg/mL) was added per well at 37 °C for 4 h, followed by shaking with DMSO (150 μL/well) for 15 min. The absorbance was measured at a wavelength of 490 nm.

#### 4.3.3. Cell Phagocytic Ability

The neutral red uptake assay was used to detect the phagocytic capacity of macrophages. After incubating with test solution for 24 h, the neutral red solution was prepared, and 20 μL of it was added to each well at 37 °C for 2 h, washed with PBS, and then we added 200 μL/well of prechilled cell lysate, and it was shaken for 10 min. The OD of each sample was observed to be 540 nm. We calculated the phagocytic viability of the cells in the administration group with reference to the cell viability of the blank group.

#### 4.3.4. ELISA

The concentrations of cytokines were measured via ELISA assays. The cytokine concentrations were determined using the ELISA kits following the manufacturer’s instructions. Briefly, 96 wells of microtiter plates coated with specific antibodies captured IL-6, IL-10, IL-12, or TNF-α in the culture-cell-free supernatants of the respectively treated samples. Then, we added the second layer of antibodies. Absorbance at 450 nm was measured by a full-wavelength microplate reader (MD Electronics, USA) for determining the cytokine concentrations.

#### 4.3.5. qRT-PCR

Real-time fluorescence quantitative PCR was used for detection. The mRNA to be detected included IL-6, IL-12, TNF-α, and IL-10. Total RNA was extracted from cells using conventional protocols. The RNA concentration measured by the absorbance ratio (A260/A280) was in the range of 1.8–2.0, which means that the RNA had good quality [56]. Subsequently, 1 μg of total RNA was reverse-transcribed to cDNA (50 °C for 15 min, 85 °C for 5 min, and 4 °C for 5 min) according to the 2 × SYBR Green qPCR Mix, and the quantitative real-time PCR (qRT-PCR) was performed using 2 × SYBR Green qPCR Mix. β-actin was chosen as the housekeeping gene. qRT-PCR was performed as follows: pre-denaturing at 95 °C for 5 min followed by 40 cycles of 95 °C for 15 s, and 60 °C for 60 s [57]. PCR primer sequences are shown in Table 2, and the 2^−ΔΔCt^ method was used to calculate the mRNA expression level of each target gene.

#### 4.3.6. Flow Cytometry

Flow cytometry was used to detect the expression of surface markers on stimulated cells. We harvested RAW264.7 cells and washed them with ice-cold PBS buffer (2% FBS and 0.1% sodium azide) before phycoerythrin (PE)-conjugated anti-CD68 antibody was added. Samples were incubated for 30 min in darkness at 4 °C and then fixed with 2% paraformaldehyde. Permeabilization wash buffer was added to break the cell membranes at room temperature. The sample with the presence of fluorescein isothiocyanate (FITC)-conjugated anti-CD206 antibody was incubated in the dark at 4 °C for 30 min, further washed, and resuspended in PBS containing 0.1% BSA. After pipetting and mixing, the surface markers were detected via a Cytomics FC 500 flow cytometer (Beckman Coulter, Brea, CA, USA).

#### 4.3.7. Western Blot

Total cell protein from harvested cells was extracted by conventional methods. After determining the protein concentration with the BCA kit, the protein was denatured and subjected to SDS-PAGE electrophoresis followed by membrane transfer. After blocking with 5% skimmed milk powder in room temperature and incubating with primary antibodies (Hes1, Jagged1, and Notch1, dilution ratios were all 1:500), TBST was used to wash the membrane, which further developed and was subsequently exposed by adding electrochemiluminescence reagent (Sparkjade, Jinan, China). The images were analyzed using the gel imaging system.

#### 4.3.8. Statistical Analysis

Results are expressed as mean ± standard deviation (x ± SD). One-way analysis of variance (ANOVA) was used for comparison between multiple groups. GraphPad Prism5 software was used for statistical analysis and graphing. *p* < 0.05 was considered to be statistically significant.

## 5. Conclusions

In conclusion, PCP-1C can promote the polarization of macrophage M1 by activating the Notch signaling pathway, suggesting that PCP-1C has the ability to further exhibit anti-tumor activity in vivo through the regulation of the immune response. The precise role of PCP-1C in vivo needs further investigation, which might encourage the exploration of more effective drugs and methods used in the immunotherapy of cancer and infectious diseases.

## Figures and Tables

**Figure 1 molecules-28-02140-f001:**
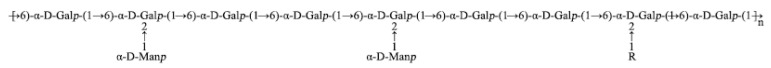
Predicted structure of the PCP-1C repeat unit. (R indicates the following fragments: T-α-D-Manp, T-α-L-Fucp, T-β-D-Glcp, 1,3-α-L-Fucp, 1,6-β-D-Glcp, 1,3-β-D-Glcp, and 1,4-β-D-Glcp residues).

**Figure 2 molecules-28-02140-f002:**
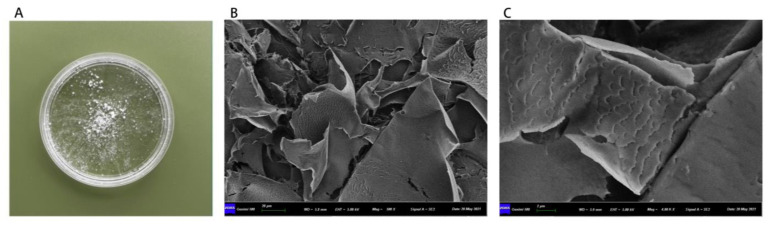
The appearance of PCP-1C: (**A**) the macroscopic morphology of PCP-1C; (**B**,**C**) SEM observation of PCP-1C using SEM at the magnification of 4000 times and 40,000 times.

**Figure 3 molecules-28-02140-f003:**
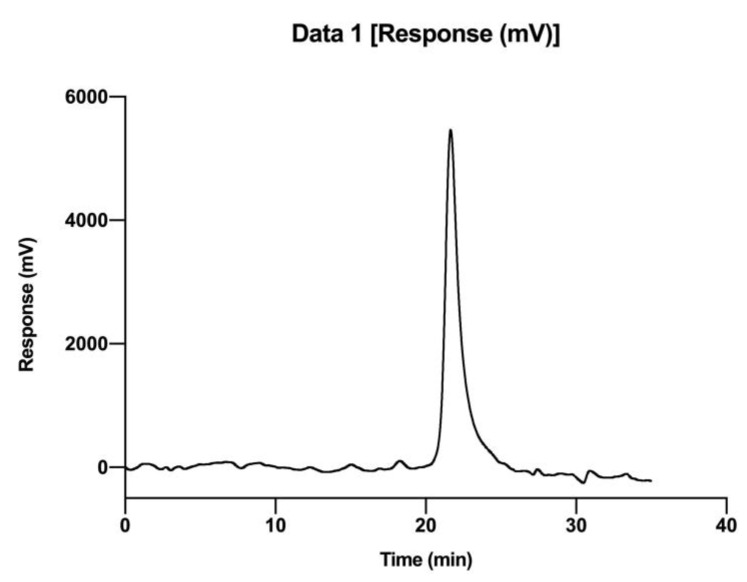
High-performance gel permeation chromatography (HPGPC) spectrum of PCP-1C.

**Figure 4 molecules-28-02140-f004:**
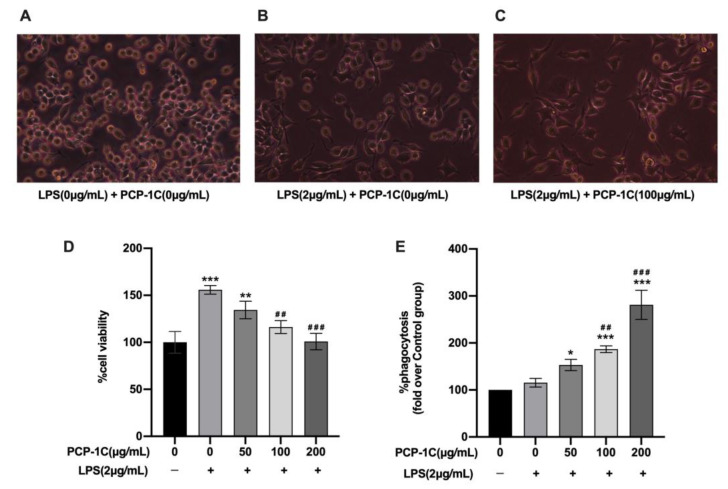
The morphology of RAW264.7 cells induced by PCP-1C and/or LPS: (**A**) macrophages cultured with blank medium; (**B**) macrophages induced by 2 μg/mL LPS for 24 h; (**C**) macrophages stimulated by 100 μg/mL PCP-1C and 2 μg/mL LPS for 24 h; viability of macrophage (**D**); and the capability of macrophages to phagocytose neutral red (**E**) after stimulation with 50 μg/mL, 100 μg/mL, and 200 μg/mL PCP-1C and/or 2 μg/mL LPS for 24 h. Cell viability (%) = (OD value of the experimental group–OD value of the blank group)**/**(OD value of the control group–OD value of the blank group) × 100%. Phagocytic ability (%) = OD value of the experimental group/OD value of the blank group × 100%. * *p* < 0.05, ** *p* < 0.01, *** *p* < 0.001 vs. Control, ## *p* < 0.01, ### *p* < 0.001 vs. LPS; “+” indicates the cells were treated with LPS, while “−” indicates they were treated without LPS.

**Figure 5 molecules-28-02140-f005:**
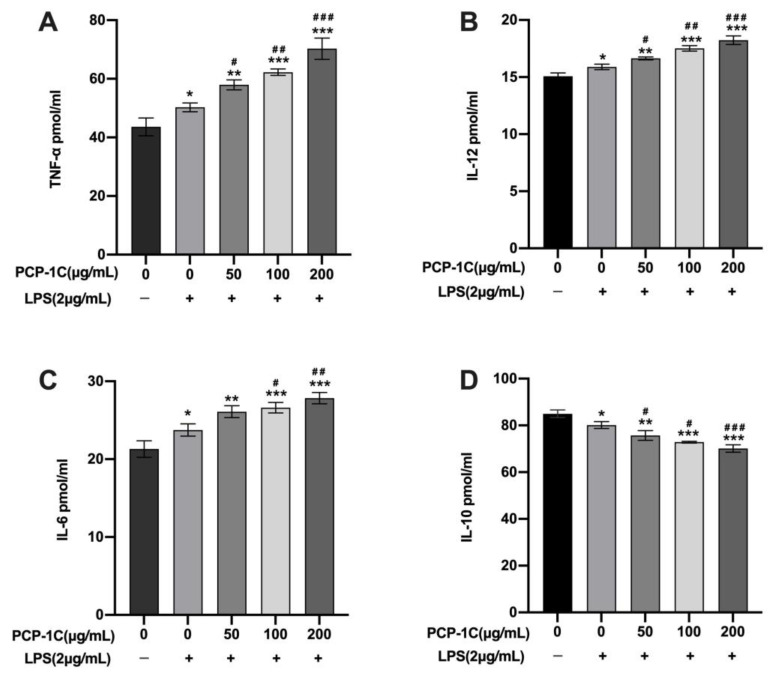
ELISA to detect macrophage cytokine levels. Macrophages were polarized at 50 μg/mL, 100 μg/mL, and 200 μg/mL PCP-1C and/or 2 μg/mL LPS for 24 h. Detected levels of cytokines TNF-α (**A**), IL-12 (**B**), IL-6 (**C**), and IL-10 (**D**) by ELISA. * *p* < 0.05, ** *p* < 0.01, *** *p* < 0.001 vs. Control, # *p* < 0.05, ## *p* < 0.01, ### *p* < 0.001 vs. LPS; “+” indicates the cells were treated with LPS, while “−” indicates they were treated without LPS.

**Figure 6 molecules-28-02140-f006:**
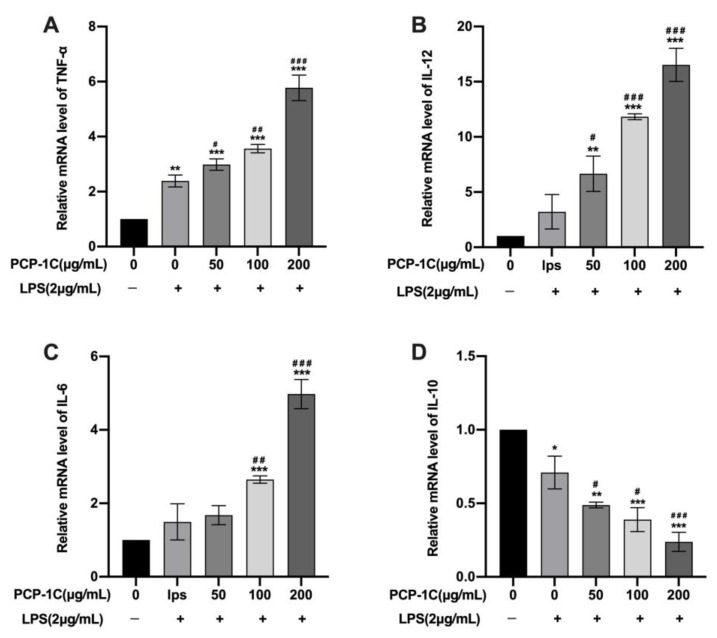
Quantitative real-time PCR (qRT-PCR) detects the mRNAs of the cytokines in macrophages. Macrophages were incubated for 24 h in the presence of 50 μg/mL, 100 μg/mL, and 200 μg/mL PCP-1C and/or 2 μg/mL LPS. mRNA expressions of cytokine levels of TNF-α (**A**), IL-12 (**B**), IL-6 (**C**), and IL-10 (**D**) were detected by qRT-PCR. * *p* < 0.05, ** *p* < 0.01, *** *p* < 0.001 vs. Control, # *p* < 0.05, ## *p* < 0.01, ### *p* < 0.001 vs. LPS; “+” indicates the cells were treated with LPS while “−” indicates they were treated without LPS.

**Figure 7 molecules-28-02140-f007:**
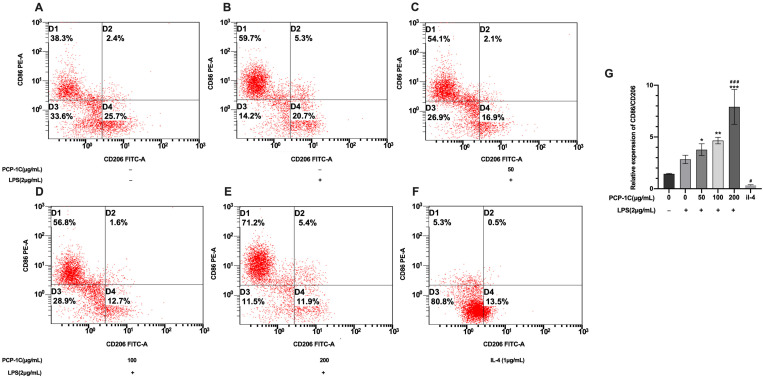
Detection of the surface markers on macrophages by flow cytometry. Macrophages were polarized at 50 μg/mL, 100 μg/mL, and 200 μg/mL PCP-1C and/or LPS for 24 h. We performed flow cytometric analysis on the polarization induced by PCP-1C and LPS in RAW264.7 cells on M1 marker CD86 and M2 marker CD206 (**A**–**F**). The change in CD86 expression/CD206 expression under the intervention of PCP-1C and LPS (**G**). * *p* < 0.05, ** *p* < 0.01, *** *p* < 0.001 vs. Control, # *p* < 0.05, ### *p* < 0.001 vs. LPS; “+” indicates the cells were treated with LPS, while “−” indicates they were treated without LPS.

**Figure 8 molecules-28-02140-f008:**
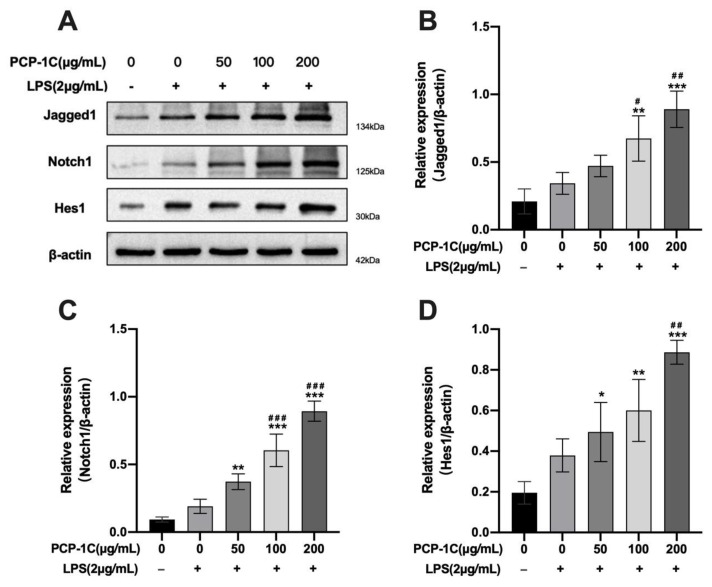
Western blot detection of Notch signaling pathway in RAW 264.7 cells. RAW 264.7 cells were stimulated with 50 μg/mL, 100 μg/mL, and 200 μg/mL PCP-1C and/or 2 μg/mL LPS for 24 h. Cell extracts were then prepared followed by Western blotting (**A**), which was performed using antibodies specific for Notch1 (**C**), Hes1 (**D**), and Jagged1 (**B**). ** p* < 0.05, ** *p* < 0.01, *** *p* < 0.001 vs. Control, # *p* < 0.05, ## *p* < 0.01, ### *p* < 0.001 vs. LPS; “+” indicates the cells were treated with LPS, while “−” indicates they were treated without LPS.

**Table 1 molecules-28-02140-t001:** Chemical compositions of PCP-1C.

Name	Sugar Content (%)	Reverse Primer	Uronic Acid Content (%)
PCP-1C	96.97 ± 1.37	0.70 ± 0.22	0.09 ± 0.02

**Table 2 molecules-28-02140-t002:** The primers used for the cytokine genes of mouse RAW264.7 cells.

	Forward Primer	Reverse Primer
IL-12	GACCTGTTTACCACTGGAACTA	GATCTGCTGATGGTTGTGATTC
IL-10	TTCTTTCAAACAAAGGACCAGC	GCAACCCAAGTAACCCTTAAAG
IL-6	CTCCCAACAGACCTGTCTATAC	CCATTGCACAACTCTTTTCTCA
TNF-α	ATGTCTCAGCCTCTTCTCATTC	GCTTGTCACTCGAATTTTGAGA
β-actin	GTCCCTCACCCTCCCAAAAG	GCTGCCTCAACACCTCAACCC

## Data Availability

The data presented in this study are available in the Results section of this article.

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
