# Peer review of "The Effect of Poria cocos Polysaccharide PCP-1C on M1 Macrophage Polarization via the Notch Signaling Pathway"

_molecules, 2023, doi:10.3390/molecules28052140_

Round 1

Reviewer 1 Report

This manuscript mainly deals with the role and mechanism of a polysaccharide in regulating macrophage activation. Generally speaking, the research is shallow and insufficient.

The main problems are as follows:

(1) The addition of IL-1b should be considered in the determination of inflammatory secretion factors.

(2) NF-kB and MAPK pathways should be considered in signal path detection

(3) The role of Notch signaling in inflammation should be fully discussed. I wonder why the author thought of this pathway in the research protocol

(4) The chemical composition and structure of the tested samples need to be further studied.

(5) Minor flaws in grammar and spelling

Author Response

Dear Reviewer,

Sincerely,

Weidong

Reviewer 2 Report

This manuscript described the effect of homogeneous Poria cocos polysaccharide PCP-1C on polarization of macrophages to M1 through Notch signaling pathway. This work has a certain contribution to the field of polysaccharide activity research. However, there are some issues to be resolved:

1.In the results section, it would be helpful to include a figure of the monosaccharide composition of PCP-1C.

2.What is the relevance between the morphological characteristics of polysaccharides and the composition of the polymer in polysaccharides?

3.Why did you not include the response of RAW264.7 cells to your extracted polysaccharide,i.e., direct treatment of RAW264.7 cells with PCP-1C alone. I suggest you should have included this in your study.

4.Figure 6 (C) (D) compared to (B) , PCP-1C only reduced the proportion of CD206+ cells, but did not increase the proportion of CD86+ cells. Is it inconsistent with the conclusion that PCP-1C can increase CD86 and decrease CD206 in macrophages? It is recommended to provide at least two additional figures regarding the results of this experiment.

5.Reference [7] Poria cocos and check all the references for the species name should be in italic fond.

6.The animal experiment will needed to emphasize the immunomodulatory activity of the extract.

Author Response

Dear reviewer,

Sincerely,

Weidong

Round 2

Reviewer 1 Report

I think it is acceptable for this revised manuscript due to the author's good responce and improvement towards the previous commments of the reviewer

Reviewer 2 Report

Overall, the justifications of the study have been articulated quite clearly, in my view; nonetheless, the study may benefit from the addition of animal based investigation.